# Experimental Investigations on the Performance of a Hollow Fiber Membrane Evaporative Cooler (HFMEC) in Hot–Dry Regions

**DOI:** 10.3390/membranes12080793

**Published:** 2022-08-18

**Authors:** Nanfeng Li, Tao Zhong, Lu Zhou, Simin Huang, Si Zeng, Caihang Liang

**Affiliations:** 1School of Mechanical and Electronic Engineering, Guilin University of Electronic Technology, Guilin 541004, China; 2Department of Science and Technology, Nanning College for Vocational Technology, Nanning 530008, China; 3Department Civil and Structural Engineering, Faculty of Architectural Engineering Design, The University of Sheffield, Sheffield City S10 2TT, UK; 4Guangdong Provincial Key Laboratory of Distributed Energy Systems, Dongguan University of Technology, Dongguan 523808, China; 5Guangxi Beitou Environmental Protection & Water Group Co., Ltd., Nanning 530025, China

**Keywords:** hollow fiber membrane, evaporative cooling, thermal comfort, wet-bulb efficiency, *COP*

## Abstract

The applicability of a hollow fiber membrane evaporative cooler in hot–dry regions was investigated by experimental studies. To better understand the actual operating environment of the hollow fiber membrane evaporative cooler, the outdoor air design conditions for summer air conditioning in five cities were simulated by an enthalpy difference laboratory. Subsequently, the effects of water and air flow rates on outlet air parameters and performance parameters were investigated by setting-up a hollow fiber membrane evaporative cooling experimental rig. It was found that the hollow fiber membrane evaporative cooler has good application prospects in hot–dry regions such as Lanzhou, Xi’an, Yinchuan, Urumqi, and Karamay. Among them, the hollow fiber membrane evaporative cooler has higher applicability in regions with higher air temperatures and lower humidity such as Urumqi and Karamay. The results indicate that the air outlet temperature and relative humidity ranged from 26.5 °C to 30.8 °C and 63.5% to 82.8%, respectively. The outlet air temperature and relative humidity of the HFMEC can meet the thermal comfort requirements of hot–dry regions in the summer at an appropriate air flow rate. The maximum air temperature drop, wet-bulb efficiency, cooling capacity, and *COP* were 7.5 °C, 62.9%, 396.4 W, and 4.81, respectively. In addition, the effect of the air flow rate on the performance parameters was more significant than that of the water flow rate.

## 1. Introduction

The consumption of fossil fuels is steadily increasing year by year as the economy grows, which results in a growing greenhouse effect [1]. In response to climate change, the Chinese government has set a clear goal to reach its carbon peak by 2030 and achieve carbon neutrality by 2060, aiming to gradually realize net-zero carbon dioxide emissions [2]. The energy consumption of the building sector in China was 1123 Mtce in 2018, accounting for approximately 20% of all commercial energy consumption, and total carbon emissions were approximately 2.2 billion tCO_2_ [3]. The energy consumption and carbon emissions of buildings will continue to grow in the coming years. The energy consumption is predicted to grow by 30% to 50% from 2015 to 2030 and by more than 50% by 2050 [4]. Air conditioning energy consumption accounts for approximately 65% of a building’s energy consumption. Therefore, air conditioning energy efficiency is a crucial initiative for energy savings and emissions reduction [5].

In recent years, the air conditioning market was still dominated by mechanical vapor compression systems [6,7,8]. However, the mechanical vapor compression system consumes a lot of electrical energy, which burdens a city’s electricity network [9]. Furthermore, converting fossil energy into electricity generates many greenhouse gases, and the leakage of refrigerants can cause ozone layer depletion and aggravate the greenhouse effect [10]. As a result, with the effects of global warming, the need for energy-saving and environmentally friendly air conditioning technology is becoming increasingly urgent.

An evaporative cooler employs the principle of heat absorption by water evaporation to cool the air stream, which is an energy-saving, environmentally friendly, and efficient air cooling method [11,12,13]. Evaporative coolers can be divided into indirect evaporative coolers [14,15] and direct evaporative coolers [16,17,18] according to the different methods of contact between the water and air stream. The outlet air humidity remains constant for an indirect evaporative cooler [15], while it increases for a direct evaporative cooler [19], which has a better cooling performance [20]. In hot–dry regions, such as Northwest China, cooling and humidifying fresh air sent into a room is necessary. Therefore, a direct evaporative cooler has better applicability [21]. However, a direct evaporative cooler demands a higher water quality, and the cooling air is entrained with water droplets [22]. The proposed evaporative cooling technology based on hollow fiber membranes overcomes these drawbacks, as the membranes with high selective permeability prevent the transmission of liquid water and microorganisms and allows for water vapor transmission [23].

Recently, some researchers have carried out a series of experimental studies and numerical simulations on the cooling performance of hollow fiber membrane evaporative coolers (HFMECs). Cui et al. conducted a parametric evaluation considering the effects of several critical parameters on a hollow fiber membrane-based semidirect evaporative cooler. The results showed that the wet-bulb efficiency of the membrane-based module could be improved to 0.73 when the inlet air velocity was 0.5 m/s [22]. Further, they investigated the air treatment performance of a membrane-based semidirect evaporative cooler with internal baffles by combining experimental studies and numerical simulations. The results show that the wet-bulb efficiency increased by 32% by setting the baffle [23]. Cheng et al., in order to avoid the flow channel or shielding of adjacent fibers to allow maximum contact between the air stream and the fiber, proposed a spindle-shaped fiber bundle membrane module. The cooling performance was investigated experimentally, and two sets of experimentally derived nondimensionless heat and mass transfer correlations were summarized [24]. They then investigated the performance parameters of a novel evaporative cooling system utilizing polymer hollow fiber spindles under different operating conditions. The results show that the proposed system significantly improved the heat and mass transfer performance compared with other designs [25]. Englart et al. discussed the thermal efficiency of a membrane module for the semidirect evaporative air cooling process by numerical simulation and proposed a method for determining the unit cooling indicator and the energy efficiency ratio [26]. Jradi et al. proposed an innovative hollow fiber membrane core to provide thermal comfort and humidity control, which consisted of 12 fiber bundles, each bundle with 1000 packed polypropylene hollow fibers. The results show that the cooling core provided a maximum cooling capacity of 502 W with 85% wet-bulb efficiency [27].

The existing research on HFMECs mainly focuses on cooling performance. However, there is less research on the applicability of HFMECs based on the meteorological parameters of different cities in hot–dry regions such as Northwest China. Therefore, the summer meteorological parameters of five typical cities in Northwest China were simulated in an enthalpy difference laboratory, and the applicability of hollow fiber membrane evaporative coolers was investigated. The effects of water and air flow rates on the outlet air parameters, air temperature drop, wet-bulb efficiency, cooling capacity, and *COP* were investigated. The results provide theoretical support for the optimal design and control strategy of HFMECs in hot–dry regions of Northwest China.

## 2. Experimental Setup

A schematic diagram of the hollow fiber membrane evaporative cooling system is shown in Figure 1. The complete system includes a hollow fiber membrane contactor, a fan, a pump, and a water tank. The circulation power of the water and air streams is provided by the water pump and fan, respectively. The water stream is pumped from the tank into the contactor and flows in the hollow fiber membrane tube. Similarly, the air stream is sent into the contactor by the fan and flows in a crossed state with the water stream on the shell side. Since there is a temperature difference and water vapor partial pressure difference between the two sides of the membrane, the water is evaporated and absorbs the sensible heat from the air stream. Therefore, the air temperature decreases while the air humidity increases.

The experimental setup of the hollow fiber membrane evaporative cooling system is shown in Figure 2. As seen from Figure 2a, a water pump (6 m head, 46 W rating) was set at the outlet of the water tank (70 L). Meanwhile, a flow meter (LZB-10, accuracy ± 1.5%) and a valve were set between the pump and the membrane module, and both of them controlled the water flow rate. The hollow fiber membrane was filled with water during the whole experimental process. Throughout the experiment, the hollow fiber membrane was filled with water. Temperature sensors (Pt100, accuracy ± 0.15 °C) were located at the inlet and outlet of the water and air streams to measure temperature changes. The air flow rate into the membrane module was controlled by varying the input voltage of the centrifugal fan (80 W rating). An anemometer (Testo 425, accuracy ± 0.03 m/s) and a humidity sensor (Center 313, accuracy ± 2.5%) were set-up at the inlet and outlet of the air channel to measure the air velocity and humidity, respectively. The structure of the membrane module is shown in Figure 2b, which was composed of several porous polymer PVDF membranes in a staggered arrangement, and its parameters are shown in Table 1. The composite membrane employed in the HFMEC was composed of two layers: a PVDF porous layer and a PVAL surface layer, and the permeability and mechanical strength were determined by the two layers together. The application of the HFMEC in Northwest China required simulating the outdoor air temperature and humidity conditions of each city and examining the variation pattern of the evaporative cooler’s performance parameters (such as air temperature drop, wet-bulb efficiency, cooling capacity, and *COP*). The enthalpy difference laboratory included two parts: the lateral chamber and the interior side, which can simulate various temperature and humidity environmental conditions. Therefore, this experiment was conducted in an enthalpy difference laboratory.

## 3. Data Reduction

### 3.1. Performance Parameters

The cooling efficiency of the evaporative cooler can be expressed by the wet-bulb efficiency (*ε*_wb_), which is defined as the ratio of the air temperature drop to the maximum possible temperature drop [28].
(1)$$\varepsilon_{\mathrm{wb}}=\frac{T_{a,\mathrm{in}}-T_{a,\mathrm{out}}}{T_{a,\mathrm{in}}-T_{a,\mathrm{wb}}}$$
where *T* is temperature (°C); subscripts “a”, “in”, “out”, and “wb” are the air stream, inlet, outlet, and wet-bulb, respectively.

The air temperature drop (Δ*T*_a,drop_) is defined as the temperature difference between the inlet and outlet of the air stream.
(2)$$\Delta T_{a,\mathrm{drop}}=T_{a,\mathrm{in}}-T_{a,\mathrm{out}}$$

The cooling capacity (*Q*_cooling_) of the HFMEC is defined as the energy change of the air stream.
(3)$$Q_{\mathrm{cooling}}=\rho_{a}V_{a}c_{p,a}\left(T_{a,\mathrm{in}}-T_{a,\mathrm{out}}\right)$$
where *c*_p,a_ is the specific heat capacity of the air stream (kJ/kg·K^−1^).

The *COP* is a key index to evaluate the performance of the evaporative cooler, which can be defined as:(4)$$COP=\frac{Q_{\mathrm{cooling}}}{W_{\mathrm{fan}}+W_{\mathrm{pump}}}$$
where *W* is the power (W).

### 3.2. Uncertainty Analysis

All the instruments were calibrated to reduce errors. The error propagation computation is required for indirect measurement parameters, and the calculation formula can be expressed as:(5)$$\Delta y={\left[{\left(\frac{\partial f}{\partial x_{1}}\right)}^{2}{\left(\Delta x_{1}\right)}^{2}+{\left(\frac{\partial f}{\partial x_{2}}\right)}^{2}{\left(\Delta x_{2}\right)}^{2}+\cdots +{\left(\frac{\partial f}{\partial x_{n}}\right)}^{2}{\left(\Delta x_{n}\right)}^{2}\right]}^{\;}$$
(6)$$\frac{\Delta y}{y}=\left[{\left(\frac{\partial f}{\partial x_{1}}\right)}^{2}{\left(\frac{\Delta x_{1}}{y}\right)}^{2}+{\left(\frac{\partial f}{\partial x_{2}}\right)}^{2}{\left(\frac{\Delta x_{2}}{y}\right)}^{2}+\cdots +{\left(\frac{\partial f}{\partial x_{n}}\right)}^{2}{\left(\frac{\Delta x_{n}}{y}\right)}^{2}\right]$$
where *f* is the function constituted by the independent variables; Δ*y* is the absolute error of the function; *x*_1_, *x*_2_, …, *x*_n_ are the independent variables constituting the function; Δ*x*_1_, Δ*x*_2_, …, Δ*x*_n_ are the absolute errors of the independent variables in the measurement process; Δ*y*/*y* is the relative error of the function constituted by the independent variables.

According to the preceding equations, the uncertainty of the wet-bulb efficiency, cooling capacity, and *COP* were 5.4%, 5.9%, and 5.3%, respectively.

## 4. Results and Discussion

### 4.1. The Effect of Inlet Air Parameters on the Performance of the HFMEC

#### 4.1.1. Inlet Air Temperature

Figure 3 shows the effect of the inlet air temperature on the performance of the HFMEC. The water and air flow rates were set to 40 L/h and 100 m^3^/h, respectively. Meanwhile, the inlet air temperature and water temperature were set to 33 °C and 23 °C, respectively. As seen in Figure 3a, the air temperature drop and wet-bulb efficiency increased from 4.22 °C and 45.4% to 6.48 °C and 53.2% as the inlet air temperature increased from 31 °C to 35 °C, respectively. As seen in Figure 3b, the cooling capacity and *COP* increased from 137.1 W and 1.91 to 210.5 W and 2.93 as the inlet air temperature increased from 31 °C to 35 °C, respectively. As the inlet air temperature rose, the temperature difference between the two sides of the membrane increased as did the heat transfer driving force of the membrane module. Therefore, the air temperature drop, wet-bulb efficiency, cooling capacity, and *COP* increased as the inlet air temperature increased.

#### 4.1.2. Inlet Air Relative Humidity

Figure 4 shows the effect of the inlet air relative humidity on the performance of the HFMEC. The water and air flow rates were set to 40 L/h and 100 m^3^/h, respectively. Meanwhile, the inlet air temperature and water temperature were set to 33 °C and 23 °C, respectively. As seen in Figure 4a, the air temperature drop decreased from 5.81 °C to 4.6 °C, while the wet-bulb efficiency increased from 44.6% to 53.4% as the inlet air relative humidity increased from 30% to 50%, respectively. As seen in Figure 4b, the cooling capacity and *COP* decreased from 188.8 W and 2.63 to 146.5 W and 2.08 as the inlet air relative humidity increased from 30% to 50%, respectively. This was because the water vapor partial pressure difference between the two sides of the membrane decreased as the relative humidity of the inlet air increased, and the mass transfer driving force decreased. Meanwhile, the water evaporation rate decreased, which resulted in less heat being removed from the air stream by water evaporation. Therefore, the air temperature dropped, and the cooling capacity and *COP* decreased as the inlet air relative humidity increased. In addition, the air wet-bulb temperature increased as the relative humidity increased. Therefore, according to Equation (1), the wet-bulb efficiency increases with increasing relative humidity in the inlet air.

### 4.2. The Effect of the Flow Rate on the Air Outlet Parameters

The cooling performance of the HFMEC was investigated by simulating the outdoor air design conditions for summer air conditioning of five typical cities in Northwest China with an enthalpy difference laboratory. The outdoor air design conditions for summer air conditioning in each city are shown in Table 2. The rated inlet temperature of the cooling water stream was set to 23 °C, and the rated flow rate of the water and air streams were 60 L/H and 100 m^3^/h, respectively. The experiments were conducted according to the set operation conditions to investigate the cooling performance and outlet air parameters of the HFMEC under different water and air flow rates. The acceptable temperature range for residents of naturally ventilated buildings is 25.0 °C to 31.6 °C, while for residents of air-conditioned buildings it is 25.1 °C to 30.3 °C. When the relative humidity of the air is between 30% and 70%, there is no significant effect on thermal comfort.

#### 4.2.1. Air Flow Rate

Figure 5 shows the air outlet parameters of the HFMEC in each city under different air flow rates (*V*_a_). The *V*_a_ increased from 25 m^3^/h to 200 m^3^/h (25 m^3^/h, 50 m^3^/h, 100 m^3^/h, 150 m^3^/h, and 200 m^3^/h along the direction of the arrows in the figure, respectively). Air velocity increased with the flow rate, reducing the time for heat-mass exchange between the air and water streams within the module, resulting in inadequate heat and mass exchange. Further, the values for *T*_a,out_ in Lanzhou, Xi’an, Yinchuan, Urumqi, and Karamay increased from 26.3 °C, 29.7 °C, 26.7 °C, 27.1 °C, and 28.9 °C to 27.4 °C, 30.8 °C, 27.9 °C, 29.1 °C, and 30.6 °C as the *V*_a_ increased from 25 m^3^/h to 200 m^3^/h, respectively. Meanwhile, the values for *RH*_a,out_ in Lanzhou, Xi’an, Yinchuan, Urumqi, and Karamay decreased from 80.6%, 82.8%, 81.5%, 80.3%, and 75.5% to 70.5%, 75%, 72.6%, 69.1%, and 63.5% as the *V*_a_ increased from 25 m^3^/h to 200 m^3^/h, respectively.

#### 4.2.2. Water Flow Rate

Figure 6 shows the air outlet parameters of the HFMEC in each city under different water flow rates (*V*_w_). The *V*_w_ increases from 40 L/h to 80 L/h (40 L/h, 50 L/h, 60 L/h, 70 L/h, and 80 L/h along the direction of the arrows in the figure, respectively). The convective heat transfer coefficient on the water side increased as the flow rate increased. Therefore, the water stream took more heat away through the membrane, and the outlet air temperature (*T*_a,out_) decreased accordingly. However, the total mass transfer coefficient was mainly determined by the air side, while the *V*_w_ change had little effect on the mass transfer, and the air outlet relative humidity (*RH*_a,out_) increase was mainly caused by the air temperature decrease. Further, the values for *T*_a,out_ in Lanzhou, Xi’an, Yinchuan, Urumqi, and Karamay decreased from 27.1 °C, 30.8 °C, 27.6 °C, 28.7 °C, and 30.3 °C to 26.5 °C, 30.1 °C, 26.8 °C, 27.8 °C, and 29.7 °C as the *V*_w_ increased from 40 L/h to 80 L/h, respectively. Meanwhile, the values of *RH*_a,out_ in Lanzhou, Xi’an, Yinchuan, Urumqi, and Karamay increased from 72.5%, 75.8%, 75.4%, 70.5%, and 66.1% to 76.1%, 80.3%, 79.3%, 75.5%, and 70.5% as the *V*_w_ increased from 40 L/h to 80 L/h, respectively.

### 4.3. Adaptability Analysis of the HFMEC in each City

#### 4.3.1. Effect of the Flow Rate on the Air Temperature Drop

Figure 7 shows the effect of the flow rate on the air temperature drop in each city. As seen in Figure 7a, the air temperature drop in Lanzhou, Xi’an, Yinchuan, Urumqi, and Karamay decreased from 6 °C, 5.2 °C, 4.6 °C, 6.4 °C, and 7.5 °C to 4.6 °C, 4.4 °C, 3.5 °C, 4.7 °C, and 6.1 °C as the air flow rate increased from 25 m^3^/h to 200 m^3^/h, respectively. Further, their values decreased by 23.3%, 15.4%, 23.9%, 26.5%, and 18.7%, respectively. The convective heat transfer coefficient on the air side increased as the air flow rate increased; therefore, more heat is carried away through the membrane by the water stream. However, the heat transfer capacity of the membrane was limited. Therefore, the heat removed by water per unit volume of air decreased as the air flow rate increased, and the air temperature drop decreased accordingly.

As seen in Figure 7b, the air temperature drop in Lanzhou, Xi’an, Yinchuan, Urumqi, and Karamay increased from 4.8 °C, 4.3 °C, 3.6 °C, 5.1 °C, and 6.5 °C to 5.5 °C, 5.0 °C, 4.3 °C, 5.8 °C, and 7.2 °C as the water flow rate increased from 40 L/h to 80 L/h, respectively. Further, their values increased by 14.6%, 16.3%, 18.6%, 13.3%, and 11.0%, respectively. This was because the convective heat transfer coefficient on the water side increased with the flow rate; therefore, the heat removed by the water stream increased. Thus, the air temperature drop increased.

According to Figure 7, it can be seen that the air temperature drop in the HFMEC under the climatic conditions of the five cities studied is from largest to smallest: Karamay, Urumqi, Lanzhou, Xi’an, and Yinchuan. This means that the HFMEC had higher air temperature drops in areas with high dry bulb temperatures and low humidity.

#### 4.3.2. Effect of the Flow Rate on the Wet-Bulb Efficiency

Figure 8 shows the effect of the flow rate on the wet-bulb efficiency in each city. As seen from Figure 8a, the wet-bulb efficiency of Lanzhou, Xi’an, Yinchuan, Urumqi, and Karamay decreased from 59.3%, 63.3%, 52%, 58.8%, and 62.9% to 46.4%, 53.5%, 38.9%, 42.6%, and 49.3% as the air flow rate increased from 25 m^3^/h to 200 m^3^/h, respectively. Further, their values decreased by 12.9%, 9.7%, 13.2%, 16.2%, and 13.6%, respectively. According to Equation (1), it can be found that the air temperature drop decreased as the air flow rate increased. Therefore, the wet-bulb efficiency had a similar trend as the air temperature drop. Meanwhile, it can be seen from Figure 8b that the wet-bulb efficiency of Lanzhou, Xi’an, Yinchuan, Urumqi, and Karamay increased from 48.9%, 53.5%, 39.9%, 46.4%, and 53.1% to 55.4%, 59.4%, 48.8%, 52.5%, and 58.9% as the water flow rate increased from 40 L/h to 80 L/h, respectively. Further, their values increased by 6.6%, 5.8%, 8.9%, 6.0%, and 5.8%, respectively.

According to Figure 8, it can be found that the wet-bulb efficiency of the HFMEC under the climatic conditions of the five cities studied is from largest to smallest: Xi’an, Karamay, Lanzhou, Urumqi, and Yinchuan.

#### 4.3.3. Effect of the Flow Rate on the Cooling Capacity

Figure 9 shows the effect of the flow rate on the cooling capacity in each city. As seen in Figure 9a, the cooling capacity of Lanzhou, Xi’an, Yinchuan, Urumqi, and Karamay increased from 48.7 W, 42.2 W, 37.4 W, 51.9 W, and 60.9 W to 298.9 W, 285.9 W, 227.4 W, 305.4 W, and 396.4 W as the air flow rate increased from 25 m^3^/h to 200 m^3^/h, respectively. Further, their values increased by 250.2 W, 243.7 W, 190.1 W, 253.4 W, and 335.5 W, respectively. On the one hand, the air temperature drop decreased as the air flow rate increased; on the other hand, the air-side convective heat and mass transfer coefficient increased with the air flow rate, which was the key factor affecting the cooling capacity of the HFMEC. Therefore, the cooling capacity increased with the increase in the air flow rate.

As seen in Figure 9b, the cooling capacity of Lanzhou, Xi’an, Yinchuan, Urumqi, and Karamay increased from 155.9 W, 139.7 W, 116.9 W, 165.7 W, and 209.6 W to 178.7 W, 162.4 W, 138.7 W, 187.8 W, and 232.6 W as the water flow rate increased from 40 L/h to 80 L/h, respectively. Further, their values increased by 22.7 W, 22.7 W, 21.8 W, 22.1 W, and 23.1 W, respectively. This was because the convective heat transfer coefficient on the water side increased as the water flow rate increased; therefore, the cooling capacity of the HFMEC increased.

According to Figure 9, it can be seen that the wet-bulb efficiency of the HFMEC under the climatic conditions of the five cities studied is from largest to smallest: Karamay, Urumqi, Lanzhou, Xi’an, and Yinchuan. This means that the HFMEC had a higher cooling capacity in areas with high dry bulb temperatures and low humidity. Additionally, increasing the flow rate of the water had no significant effect on the mass transfer performance of the HFMEC. The effect of the water flow rate on the cooling capacity of the HFMEC was therefore smaller than that of the air flow rate.

#### 4.3.4. Effect of the Flow Rate on the COP

Figure 10 shows the effect of the flow rate on the *COP* in each city. As seen in Figure 10a, the *COP* of Lanzhou, Xi’an, Yinchuan, Urumqi, and Karamay increased from 0.77, 0.67, 0.59, 0.82, and 0.96 to 3.63, 3.47, 2.76, 3.70, and 4.81 as the air flow rate increased from 25 m^3^/h to 200 m^3^/h, respectively. Further, their values increased by 2.86, 2.81, 2.17, 2.89, and 3.85, respectively. The HFMEC’s cooling capacity and fan power increased as the air flow rate increased. The effect of the increasing air flow rate on the cooling capacity, however, was more significant. Therefore, the COP increased as the air flow rate increased.

As seen in Figure 10b, the *COP* of Lanzhou, Xi’an, Yinchuan, Urumqi, and Karamay increased from 2.17, 1.95, 1.63, 2.31, and 2.92 to 2.49, 2.26, 1.93, 2.61, and 3.24 as the water flow rate increased from 40 L/h to 80 L/h, respectively. Further, their values increased by 0.32, 0.32, 0.30, 0.31, and 0.32, respectively. The water flow rate was controlled by the valve opening. Therefore, the effect of the water flow rate’s variation on the system’s power was negligible.

According to Figure 10, it can be seen that the wet-bulb efficiency of the HFMEC under the climatic conditions of the five cities studied is from largest to smallest: Karamay, Urumqi, Lanzhou, Xi’an, and Yinchuan. The *COP* increase was mainly caused by the cooling capacity increase. There was a similar tendency between the *COP* and the cooling capacity, and the air flow rate had a greater effect on the *COP* than the water flow rate.

## 5. Conclusions

To investigate the performance of an HFMEC in hot–dry regions, such as Northwest China, five typical cities were simulated in an enthalpy difference laboratory. We investigated the effect of flow rates on the air outlet parameters, air temperature drop, wet-bulb efficiency, cooling capacity, and *COP* of the HFMEC. Based on the findings, the following conclusions can be drawn:(1)The HFMEC performed better under the climates of Urumqi and Karamay, which means it provides a higher level of performance in areas with higher air temperatures or lower humidity levels. Therefore, it has good applicability in hot–dry regions such as Northwest China;(2)The air outlet temperature of the HFMEC varied from 26.5 °C to 30.8 °C under the climatic conditions of Lanzhou, Xi’an, Yinchuan, Urumqi, and Karamay. For regions with high air temperatures, such as Xi’an and Karamay, employing lower inlet water temperatures or a membrane with better thermal conductivity can reduce the air outlet temperature and improve the applicability of HFMECs in hot regions;(3)The air outlet relative humidity of the HFMEC ranged from 63.5% to 82.8% in each city. The air outlet relative humidity in Lanzhou, Urumqi, and Karamay satisfied the indoor thermal comfort requirements when the air flow rate was 200 m^3^/h. In addition, the air outlet relative humidity of the module in Xi’an and Yinchuan required mixing with the fresh outdoor air to meet the indoor thermal comfort requirements. Hence, using a membrane with a lower diffusivity can reduce the relative humidity at the air outlet in order to improve the applicability of HFMECs;(4)The effect of the air flow rate on the performance parameters was more significant than that of the water flow rate. The maximum cooling capacity and *COP* of the HFMEC could reach 396.4 W and 4.81, respectively. In addition, the maximum air temperature drop and wet-bulb efficiency could reach 7.5 °C and 62.9%, respectively.

## Figures and Tables

**Figure 1 membranes-12-00793-f001:**
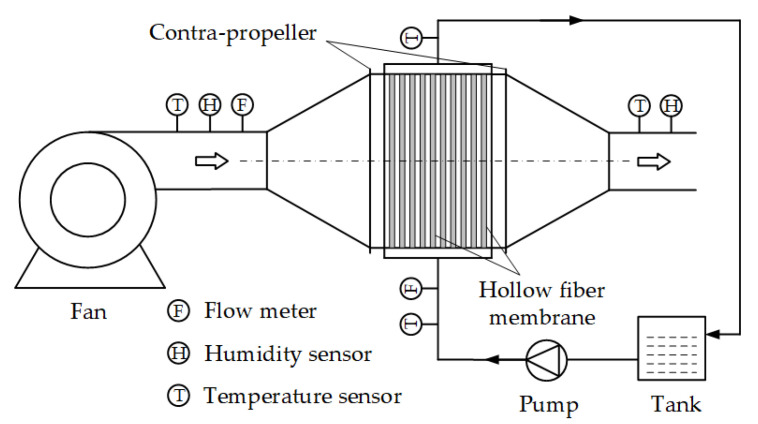
Schematic diagram of the hollow fiber membrane evaporative cooling system.

**Figure 2 membranes-12-00793-f002:**
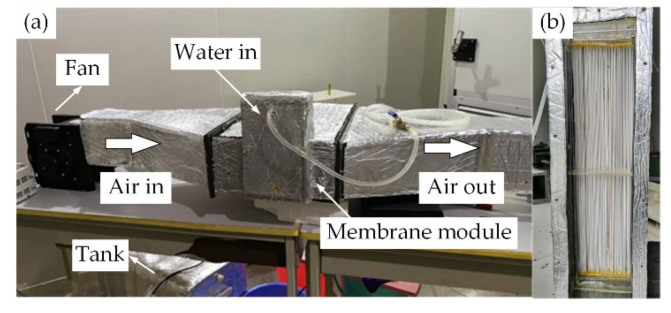
Experimental setup of the hollow fiber membrane evaporative cooling system: (**a**) hollow fiber membrane evaporative cooling system; (**b**) membrane module.

**Figure 3 membranes-12-00793-f003:**
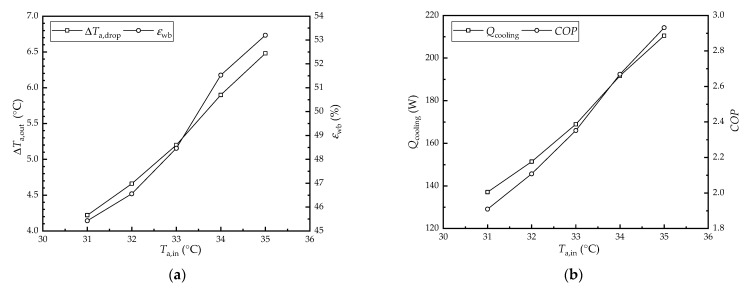
The effect of the inlet air temperature on the performance of the HFMEC: (**a**) air temperature drop and wet-bulb efficiency; (**b**) cooling capacity and *COP*.

**Figure 4 membranes-12-00793-f004:**
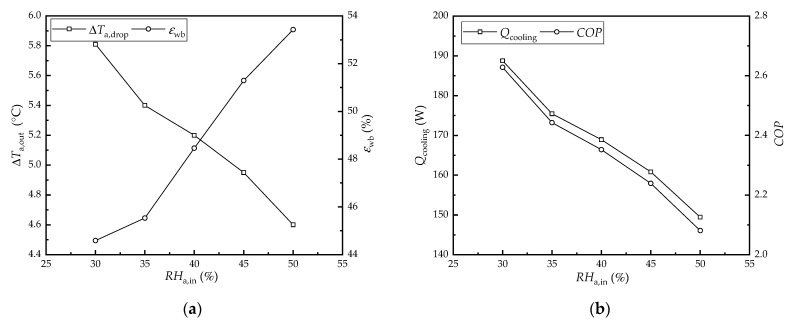
The effect of the inlet air relative humidity on the performance of the HFMEC: (**a**) air temperature drop and wet-bulb efficiency; (**b**) cooling capacity and *COP*.

**Figure 5 membranes-12-00793-f005:**
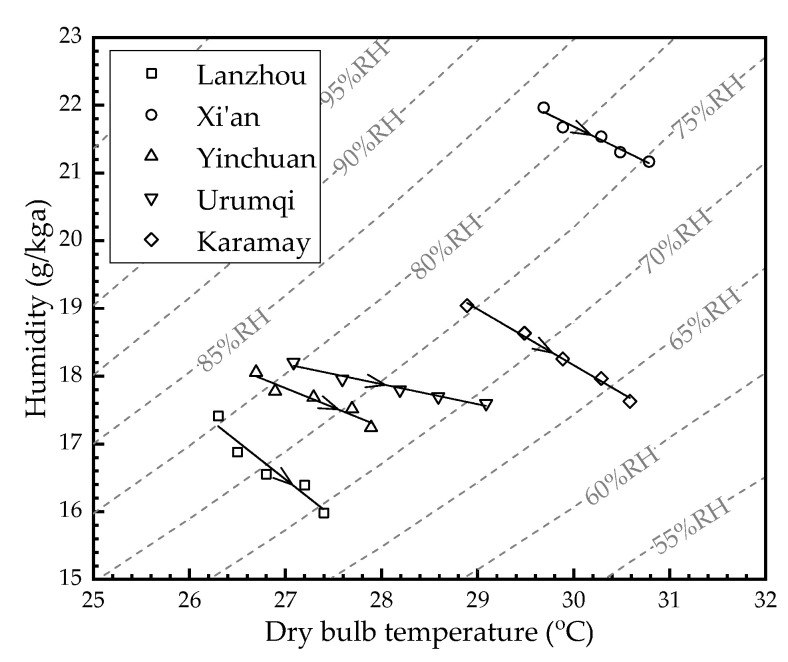
The air outlet parameters of the HFMEC in each city under different air flow rates.

**Figure 6 membranes-12-00793-f006:**
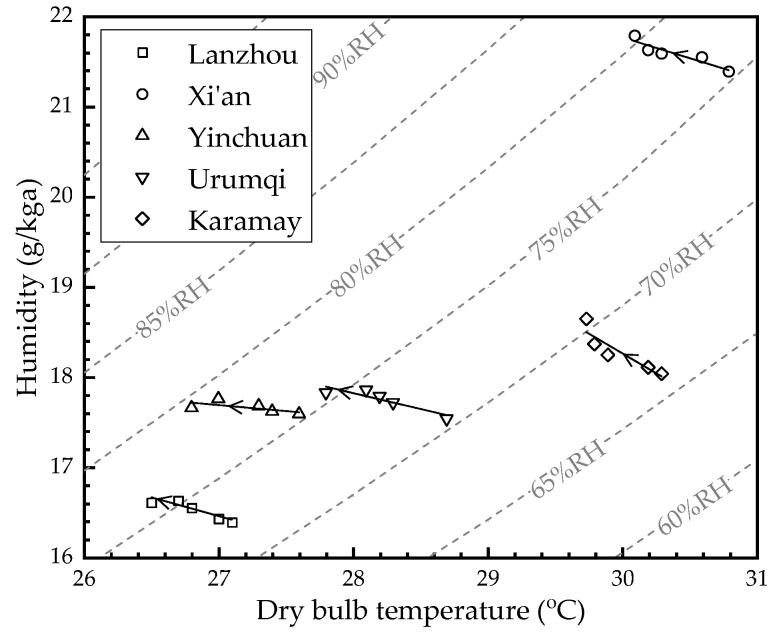
The air outlet parameters of the HFMEC in each city under different water flow rates.

**Figure 7 membranes-12-00793-f007:**
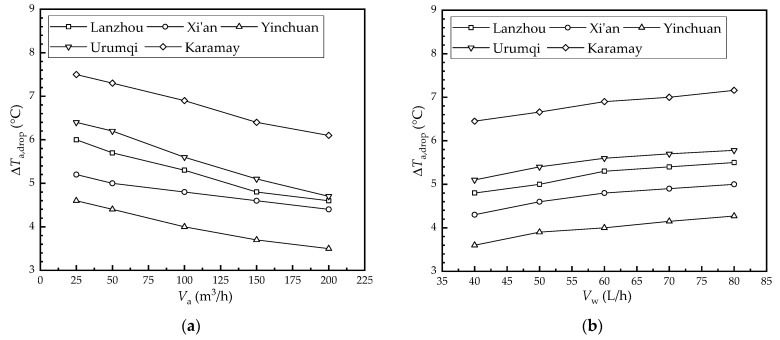
Effect of the flow rate on the air temperature drop in each city: (**a**) air flow rate; (**b**) water flow rate.

**Figure 8 membranes-12-00793-f008:**
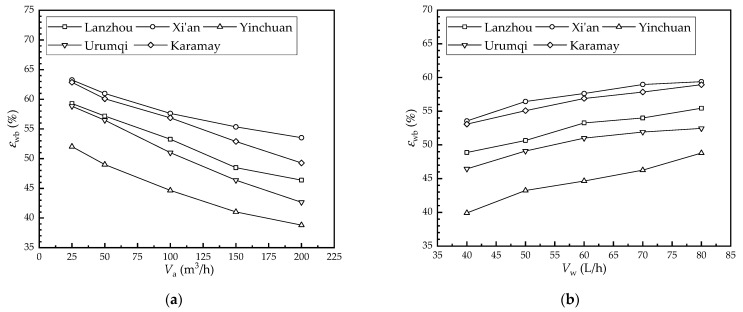
Effect of the flow rate on the wet-bulb efficiency in each city: (**a**) air flow rate; (**b**) water flow rate.

**Figure 9 membranes-12-00793-f009:**
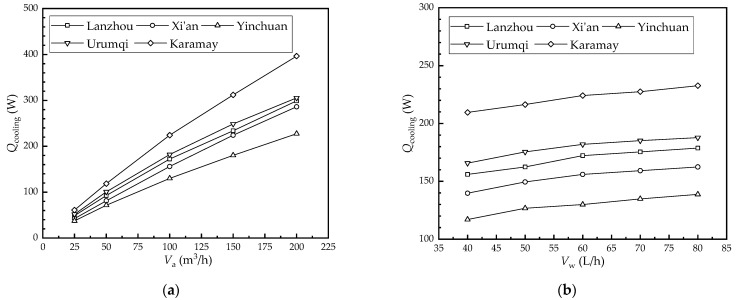
Effect of the flow rate on the cooling capacity in each city: (**a**) air flow rate; (**b**) water flow rate.

**Figure 10 membranes-12-00793-f010:**
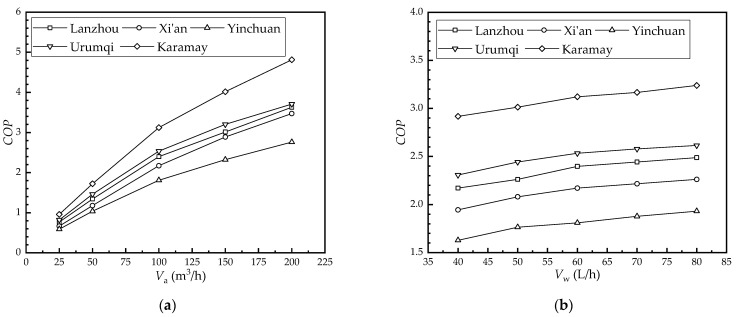
Effect of the flow rate on the *COP* in each city: (**a**) air flow rate; (**b**) water flow rate.

**Table 1 membranes-12-00793-t001:** Parameters of the hollow fiber membrane module.

Module	Value	Unit	Membrane	Value	Unit
Inner diameter	1.2	mm	Pore size	0.15	μm
Outer diameter	1.5	mm	PVAL skin layer thickness	40	μm
Tube pitch	3	mm	PVDF porous layer thickness	110	μm
Module size	500 × 150 × 150	mm	Membrane thickness	150	μm
Tube number	2500	-	Diffusivity	9 × 10^−7^	m^2^/s
Arrangement	Staggered	-	Thermal conductivity	0.17	W/(m·K)

**Table 2 membranes-12-00793-t002:** Outdoor air design conditions for summer air conditioning of each city.

City	Name	*T*_a,db_ (°C)	*T*_a,wb_ (°C)	*RH*_a,in_ (%)	*ω*_a,in_ (g/kg)
1	Lanzhou	32.1	22.1	42.7	12.8
2	Xi’an	35.1	26.8	53.1	19.0
3	Yinchuan	31.3	22.3	46.7	13.4
4	Urumqi	33.7	22.7	39.5	12.9
5	Karamay	36.7	24.6	37.8	14.7

## Data Availability

The data presented in this study are available in the article.

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
