# Peer review of "Experimental Investigations on the Performance of a Hollow Fiber Membrane Evaporative Cooler (HFMEC) in Hot–Dry Regions"

_membranes, 2022, doi:10.3390/membranes12080793_

Round 1

Reviewer 1 Report

Reducing the energy consumption of systems such as air conditioning is an important task. The method of cooling air by evaporation of water is an ancient technology. In Arab countries, a decorative element called mashrabiya was used for these purposes. The essence of this technology is to increase the evaporation area. Highly porous materials or membranes can be used for this.

In this paper, the issue of the use of membranes for air cooling is investigated. The authors focus on experimental research. However, there are a number of questions to this work.

1. It is not clear which membrane was used. Only its geometric characteristics are given in the work. While the main most interesting characteristic is the water vapor permeability coefficient.

2. 2. Section 3.2 talks about error calculations, but the error limits are not shown on the graphs.

3. The results of the experiments are natural. The authors easily explain the resulting dependencies. The nature of the obtained dependencies could be predicted even before the experiments. In the results of the work, I would like to see an assessment of the effectiveness of such a system in the premises, as it causes, at first glance, doubt. A stream cooled by several degrees is unlikely to change the indoor climate.

4. Was the fact taken into account that the water that will circulate in a closed circuit will eventually heat up under the action of the pump and, as the authors noted, the heat removed from the air. And how is it supposed to operate a possible system, since all the electrical appliances used emit more heat. Will the remote unit be used?

Author Response

Dear Professor:

       The responese to the reviewer's comments, please see the attachment. 

Reviewer 2 Report

The authors experimentally investigated hollow fiber membrane evaporative cooler in regions with higher air temperature and lower humidity. Furthermore, the air outlet temperature and relative humidity range from 26.5 °C to 30.8 °C and 63.5% to 82.8%, respectively. The outlet air temperature and relative humidity of the HFMEC were in an acceptable range.  I recommend addressing the following comments 

1. Improve the figure quality 

2. The English language needs to be improved slightly 

Author Response

Dear Professor,

Thank you for organizing the reviews and thanks to the reviewers for providing useful questions, comments, and suggestions. The response are in the attachemnt, Please see the attachment

Reviewer 3 Report

The work discusses the operation of a capillary membrane cooler—this device is used in areas with high temperatures and dry air. The research concerned a capillary air cooler of the HFMEC type. The study was out experimentally, analyzing the climatic conditions in several country regions. The test results made it possible to correct the structure's parameters to the country's dedicated climatic conditions.

The work is concise, and the conclusions are logical. The readability of the charts needs to be improved. However, the results themselves do not raise any objections.

I believe the work is attractive in terms of engineering; it can have a group of readers and meet the demand for sustainable development of the country.

I noticed the following things for improvement.

1. Page 3, row 117  "Therefore, to investigate the applicability of HFMEC in hot-dry regions, the climate of five typical cities in northwest China is simulated in an enthalpy difference laboratory." - Unclear sentence...

2.Page 8, Figure 5, The vertical axis should be on the left side for the Cartesian coordinates and the first quarter.

3. Page 9, figure 6, ... as above.

4. Page 10, Figure 7,Figure 8. The markings should have a font size equal to the text- see Figure 1.

Thanks.

Author Response

Dear Professor
    The response to reviewer‘s comments are in attachment, Please see the attachment.
